# The Fisher Dimension: Instance-Dependent Complexity for Causal Discovery

Luong Doan [1 2]   Khanh Nguyen [1 2]   Hai Nguyen [1 2]   Hung Mai [1 3]   Phong Ho [1]   Nhung Duong [1 4]   Tuan Do [1 2]

## Abstract

Classical sample complexity bounds for causal structure learning are minimax in nature, characterizing worst-case difficulty without distinguishing between easy and hard instances. We study instance-specific complexity for Markov equivalence class (MEC) recovery in linear Gaussian structural equation models. We introduce the Fisher dimension, defined as the inverse squared minimum partial correlation that must be detected to recover the MEC. We prove that the Fisher dimension governs sample complexity: it provides both a lower bound and an upper bound (tight up to logarithmic factors) for MEC recovery. A key theoretical finding is that under spectrally well-conditioned models, with bounded noise variances, bounded covariance eigenvalues, and constant-order edge coefficients, the Fisher dimension is uniformly bounded regardless of graph structure. Thus, significant instance-specific variation arises from parametric rather than structural features. Empirical validation shows strong correlation between our predictor and observed sample complexity for structured graph families.

## 1. Introduction

Causal discovery from observational data is a fundamental problem in machine learning and statistics (Spirtes et al., 2001). Given $n$ independent samples from a distribution $P$ that is Markov with respect to an unknown DAG $G^*$ over $d$ variables, the goal is to recover the Markov equivalence class (MEC) $[G^*]$, the set of DAGs encoding the same conditional independencies as $G^*$. This problem has been extensively studied (Vowels et al., 2022; Assaad et al., 2022), with classical results establishing consistency of constraint-based

algorithms (PC) and score-based methods (GES) as $n \to \infty$ under faithfulness assumptions.

The sample complexity of causal discovery has received considerable attention. For linear Gaussian DAGs with maximum in-degree $q$, recent work (Gao et al., 2022) shows that $n \asymp q \log(d/q)$ samples are minimax optimal. However, these bounds characterize worst-case complexity and do not explain why some graphs are substantially easier to learn than others. Dense graphs with many v-structures provide more orienting information than sparse chains, yet no existing theory precisely characterizes this instance-specific difficulty. We address this gap by introducing the Fisher dimension $\mathcal{F}([G]) = 1/\rho_{\min}^2$, where $\rho_{\min}$ is the smallest nonzero partial correlation that must be detected for MEC recovery. We prove that $\mathcal{F}([G])$ is both necessary and sufficient (up to $\log d$) for successful causal discovery.

### 1.1. Our contributions

First, we define the Fisher dimension $\mathcal{F}([G]) = 1/\rho_{\min}^2$ over a sufficient test set that guarantees MEC recovery. This definition is algorithm-independent and has a natural information-geometric interpretation: the Fisher dimension captures the minimum curvature (maximum flatness) among directions that distinguish $[G]$ from other MECs in the space of covariance matrices.

Second, we prove upper and lower bounds on sample complexity: $n = \Omega(\mathcal{F}([G]))$ and $n = O(\mathcal{F}([G]) \cdot \log d)$. The upper bound is achieved by PC via Fisher's z-transform analysis; the lower bound uses Le Cam's method on MECs differing by a single partial correlation. These bounds match up to $\log d$, identifying $\mathcal{F}([G])$ as the right instance-dependent complexity parameter.

Third, we establish that under spectrally well-conditioned SEMs–with bounded noise variances, bounded covariance eigenvalues, and $\Theta(1)$ edge coefficients–the Fisher dimension satisfies $\mathcal{F}([G]) = O(1)$ independently of graph structure. Chains, trees, and dense DAGs all have the same asymptotic rate under these conditions.

Fourth, we prove $\mathcal{F}([G]) = \Theta(1)$ for complete DAGs (Proposition 4.4), providing a concrete example. We characterize regimes where $\mathcal{F}([G])$ can grow with $d$, and prove a curvature characterization $\mathcal{F}([G]) \asymp \lambda_{\min}(\mathcal{C}_{[G]})^{-1}$ for

---

[1]N2TP Technology Solutions JSC, Hanoi, Vietnam [2]Phenikaa University, Hanoi, Vietnam [3]Hanoi National Economics University, Hanoi, Vietnam [4]Hanoi University of Pharmacy, Hanoi, Vietnam. Correspondence to: Luong Doan <luong.doan.wk@gmail.com>, Tuan Do <tuando7758@gmail.com>.

*Proceedings of the 43rd International Conference on Machine Learning*, Seoul, South Korea. PMLR 306, 2026. Copyright 2026 by the author(s).

chains under weak-edge scaling (Theorem 3.11), with explicit constants.

Fifth, we provide Algorithm 1 for computing $\mathcal{F}([G])$ with complexity $O(|E| \cdot d^3)$, and establish finite-sample guarantees: concentration bounds (Lemma 5.2) and $O(n^{-1/2})$ convergence (Proposition 5.3) for estimated Fisher dimension.

Sixth, we empirically validate our theory on synthetic DAGs. Across tree, chain, and Erdős–Rényi families, $\mathcal{F}([G]) \log d$ correlates strongly with empirical sample complexity, and $\mathcal{F}([G])$ outperforms simpler structural proxies for structured families.

### 1.2. Related Works

Causal discovery has been studied extensively since Spirtes et al. (2001), who introduced the Markov and faithfulness assumptions and the PC algorithm. Verma & Pearl (1990) characterized MECs via v-structures, and Chickering (2002) showed score-based methods face the same equivalence limitations. Richardson & Spirtes (2002) extended these ideas to latent variable models; see Peters et al. (2017) for a comprehensive overview.

High-dimensional consistency under beta-min conditions was established by Kalisch & Bühlmann (2007) for PC and van de Geer & Bühlmann (2013) for penalized likelihood. These results provide sufficient conditions but do not characterize how difficulty varies across MECs.

Uhler et al. (2013) studied the geometry of faithfulness, showing distributions can be arbitrarily close to violating it. Shah & Peters (2020) established hardness results for CI testing in high dimensions. Our work builds on these insights by identifying the specific partial correlations critical for distinguishing MECs.

A related line investigates identifiability beyond Markov equivalence: Peters & Bühlmann (2014) showed equal error variances enable full identifiability, extended by Wu & Drton (2023) to partial homoscedasticity. Wang & Drton (2020; 2023) established consistency for non-Gaussian models with latent confounding, and Agrawal et al. (2023) addresses nonlinear settings. We focus on instance-dependent complexity within the classical Gaussian MEC setting.

Gao et al. (2022) establishes minimax bounds for MEC recovery, showing worst-case complexity scales with global quantities. While these characterize fundamental limits, they do not explain instance-to-instance variation. We address this gap: the Fisher dimension is both necessary and sufficient (up to $\log d$) and clarifies when graph structure versus parametric properties drive sample complexity.

## 2. Preliminaries

### 2.1. Linear Gaussian Structural Equation Models

A linear Gaussian SEM over variables $X = (X_1, \ldots, X_d)$ is defined by

$$X_i = \sum_{j \in \mathrm{Pa}(i)} \beta_{ji} X_j + \varepsilon_i, \quad \varepsilon_i \sim \mathcal{N}(0, \sigma_i^2),$$

where $\mathrm{Pa}(i)$ denotes the parents of node $i$ in DAG $G$, $\beta_{ji}$ are the edge coefficients, and $\varepsilon_i$ are independent Gaussian noise terms. The joint distribution is multivariate Gaussian with covariance matrix $\Sigma = (I - B)^{-1} D (I - B)^{-\top}$, where $B$ is the matrix of edge coefficients and $D = \mathrm{diag}(\sigma_1^2, \ldots, \sigma_d^2)$.

### 2.2. Markov Equivalence Classes

Two DAGs $G_1, G_2$ are Markov equivalent if they encode the same set of conditional independencies. The equivalence class $[G]$ is characterized by having the same skeleton (undirected edges) and same v-structures (immoralities), where a v-structure is a configuration $i \to k \leftarrow j$ with $i$ and $j$ non-adjacent. The equivalence class is represented by a completed partially directed acyclic graph (CPDAG), which contains directed edges for those whose orientation is determined by v-structures and undirected edges for those that can be oriented either way.

### 2.3. Partial Correlations and Conditional Independence

For variables $X_i, X_j$ and conditioning set $S \subseteq [d] \setminus \{i, j\}$, the partial correlation is

$$\rho_{ij|S} = \mathrm{Corr}(X_i, X_j \mid X_S).$$

For Gaussian distributions, $\rho_{ij|S} = 0$ if and only if $X_i \perp\!\!\!\perp X_j \mid X_S$. This equivalence is the foundation of constraint-based causal discovery algorithms such as PC, which recover the MEC by testing conditional independencies through partial correlation tests.

### 2.4. The PC Algorithm

The PC algorithm recovers the Markov equivalence class $[G^*]$ in two phases. In the skeleton phase, it tests conditional independencies to determine which pairs of variables are adjacent. Starting from a complete undirected graph, it removes edge $(i, j)$ if it finds a conditioning set $S$ such that $X_i \perp\!\!\!\perp X_j \mid X_S$, which is equivalent to $\rho_{ij|S} = 0$ for Gaussian distributions. In the orientation phase, it identifies v-structures by finding configurations $i - k - j$ where $i$ and $j$ are non-adjacent and $k \notin S_{ij}$ (the separating set that rendered $i$ and $j$ independent), then orients as $i \to k \leftarrow j$. Additional compelled orientations are then obtained by ap-

plying Meek's orientation rules, which propagate arrowheads while avoiding directed cycles and the introduction of new unshielded colliders (Meek, 1995).

## 2.5. Statistical Manifolds and Fisher Information

For a parametric family $\{P_\theta : \theta \in \Theta \subseteq \mathbb{R}^k\}$, the *statistical manifold* $\mathcal{M}$ is the set of distributions equipped with the Fisher–Rao metric. The *Fisher information matrix* is defined as

$$I(\theta)_{ij} = \mathbb{E}_{P_\theta}\left[\frac{\partial \log p_\theta(X)}{\partial \theta_i} \cdot \frac{\partial \log p_\theta(X)}{\partial \theta_j}\right].$$

The Cramér-Rao bound states that for any unbiased estimator $\hat{\theta}$ of $\theta$, $\mathrm{Var}(\hat{\theta}) \geq I(\theta)^{-1}$, establishing Fisher information as the fundamental limit on estimation precision.

For a DAG $G$, the *causal manifold* $\mathcal{M}_G$ is the set of distributions factorizing as $p(x_1, \ldots, x_d) = \prod_{i=1}^d p(x_i \mid x_{\mathrm{Pa}(i)})$. For linear Gaussian SEMs, the parameter space is $\Theta_G = \{(\beta, \sigma^2) : \beta_{ij} \neq 0 \Rightarrow (j, i) \in E(G)\}$, encoding the sparsity pattern of the DAG.

# 3. Fisher Dimension and Main Results

## 3.1. Definition of Fisher Dimension

We define the Fisher dimension through the concept of a sufficient test set. For a DAG $G^*$, consider the set of partial correlation tests that suffice to recover $[G^*]$. We focus on the canonical sufficient test set $\mathcal{T}([G^*])$ containing, for each edge $(j, i) \in E(G^*)$, the test $(i, j, S)$ with $S = \mathrm{Pa}(i) \setminus \{j\}$ (the canonical conditioning set), and for each v-structure $i \to k \leftarrow j$, a test confirming non-adjacency of $i, j$.

**Definition 3.1** (Fisher Dimension). For a linear Gaussian SEM with true DAG $G^*$, define

$$\mathcal{F}([G^*]) := \frac{1}{\rho_{\min}^2([G^*])},$$

where

$$\rho_{\min}([G^*]) := \min\left\{|\rho_{ij|S}| : (i, j, S) \in \mathcal{T}([G^*]),\right.$$
$$\left.\rho_{ij|S} \neq 0\right\}.$$

The restriction to nonzero correlations excludes v-structure tests (where we detect $\rho = 0$); these do not affect the sample complexity scaling.

The Fisher dimension captures how difficult it is to orient the hardest edge. Large $\mathcal{F}([G])$ corresponds to small $\rho_{\min}$, meaning weak signal for edge orientation, which requires more samples. Small $\mathcal{F}([G])$ corresponds to large $\rho_{\min}$, meaning strong signal that can be detected with fewer samples.

The partial correlation $\rho_{ij|S}$ directly measures the strength of conditional dependence between $X_i$ and $X_j$ given $X_S$ in linear Gaussian models. The Fisher information for estimating $\rho_{ij|S}$ from $n$ samples is approximately $I_n(\rho_{ij|S}) \approx n$ for small $|\rho_{ij|S}|$, so detecting a partial correlation of magnitude $\rho$ requires $n \gtrsim 1/\rho^2$ samples. The Fisher dimension captures exactly this sample complexity scaling for the hardest-to-detect edge.

*Remark* 3.2 (Information-Geometric Interpretation). The partial-correlation-based definition has a natural information-geometric interpretation. With $\rho$ short for $\rho_{ij|S}$, standardizing the conditional covariance to the bivariate family $\Sigma(\rho) = \begin{pmatrix} 1 & \rho \\ \rho & 1 \end{pmatrix}$ yields the Fisher–Rao metric coefficient

$$g_{\rho\rho} = \frac{1}{2} \mathrm{tr}\big(\Sigma(\rho)^{-1}\partial_\rho\Sigma(\rho)\,\Sigma(\rho)^{-1}\partial_\rho\Sigma(\rho)\big) = \frac{1 + \rho^2}{(1 - \rho^2)^2}.$$

Thus distinguishing DAGs in $[G]$ from those outside requires detecting movement along uncertain edge directions. The Fisher dimension captures the *minimum curvature* (equivalently, maximum flatness) among these distinguishing directions–flat directions are hard to detect, requiring more samples.

*Remark* 3.3 (Algorithm-Independence). Equivalently, $\rho_{\min}([G^*])$ is the smallest nonzero partial correlation that distinguishes $[G^*]$ from some other MEC. Thus Fisher dimension is intrinsically algorithm-independent: it characterizes the underlying instance, not the behavior of any specific learning method. The PC algorithm achieves the upper bound (Theorem 3.4) by explicitly testing these correlations, while the lower bound (Theorem 3.6) applies to *any* algorithm. Therefore Fisher dimension should be interpreted as an information-theoretic instance parameter, not as the realized sample complexity of a particular algorithm.

## 3.2. Upper Bound on Sample Complexity

**Theorem 3.4** (Upper Bound). *Let $G^*$ be the true DAG over $d$ variables with Markov equivalence class $[G^*]$ and Fisher dimension $\mathcal{F}([G^*]) = 1/\rho_{\min}^2$. Assume the linear Gaussian SEM has maximum degree $\Delta$. There exists an algorithm (the PC algorithm) that, given $n$ i.i.d. samples, outputs $[\hat{G}]$ such that*

$$\mathbb{P}\left([\hat{G}] = [G^*]\right) \geq 1 - \delta,$$

*provided*

$$n \geq C \cdot \mathcal{F}([G^*]) \cdot \log(d/\delta),$$

*where $C > 0$ is a constant depending on $\Delta$ and the condition number of the covariance matrix.*

*Proof Sketch.* The PC algorithm tests conditional independencies via partial correlations. Fisher's z-transform $\hat{z}_{ij|S} = \frac{1}{2} \log \frac{1+\hat{\rho}_{ij|S}}{1-\hat{\rho}_{ij|S}}$ satisfies $\hat{z}_{ij|S} \sim \mathcal{N}(z_{ij|S}^*, (n-|S|-3)^{-1})$. To

detect partial correlations of magnitude $\geq \rho_{\min}$ with power $1-\beta$, we need $n = O(1/\rho_{\min}^2)$ per test. With $T = O(d^{2+\Delta})$ total tests, a union bound with $\alpha = \beta = \delta/(2T)$ yields the stated complexity. See Appendix B for the full proof. $\square$

*Remark* 3.5 (On the $\log d$ Factor). The sample complexity scales as $\mathcal{F}([G]) \cdot \log d$, not $\mathcal{F}([G]) \cdot d^2$. Although there are $O(d^2)$ potential edges, they share the same $n$ samples, contributing only logarithmically through the union bound. This is consistent with standard results on Gaussian graphical model learning.

### 3.3. Lower Bound on Sample Complexity

**Theorem 3.6** (Lower Bound). *Let $G_0, G_1$ be two DAGs with $[G_0] \neq [G_1]$ such that the corresponding Gaussian distributions $P_0, P_1$ differ only in a partial correlation: $\rho_{ij|S} = \rho_0$ under $P_0$ and $\rho_{ij|S} = 0$ under $P_1$. Then for any algorithm $\mathcal{A}$,*

$$\inf_{\mathcal{A}} \left[ \mathbb{P}_{P_0^n}(\mathcal{A} \neq [G_0]) + \mathbb{P}_{P_1^n}(\mathcal{A} \neq [G_1]) \right] \geq \frac{1}{2}$$

*if $n \leq c/\rho_0^2$ for a universal constant $c > 0$.*

*Proof Sketch.* We use Le Cam's two-point method. For Gaussians differing only in a partial correlation of magnitude $\rho_0$, the KL divergence satisfies $\mathrm{KL}(P_0\|P_1) = O(\rho_0^2)$. For $n$ i.i.d. samples, $\mathrm{KL}(P_0^n\|P_1^n) = n \cdot \mathrm{KL}(P_0\|P_1) \leq Cn\rho_0^2$. By Pinsker's inequality, $\mathrm{TV}(P_0^n, P_1^n) \leq \sqrt{n\rho_0^2/2}$. Le Cam's lemma then implies the sum of error probabilities is at least $1 - \mathrm{TV}(P_0^n, P_1^n) \geq 1/2$ when $n \leq c/\rho_0^2$. See Appendix C for the full proof. $\square$

**Corollary 3.7.** *For any faithful linear Gaussian SEM with DAG $G$,*
$$n = \Omega(\mathcal{F}([G]))$$
*samples are necessary for any algorithm to recover $[G]$.*

Theorems 3.4 and 3.6 together establish that $n = \Omega(\mathcal{F}([G]))$ is necessary and $n = O(\mathcal{F}([G]) \cdot \log d)$ is sufficient for learning $[G]$. The $\log d$ gap between upper and lower bounds comes from the union bound over $O(d^{2+\Delta})$ tests; closing this gap remains an open problem.

### 3.4. Curvature Characterization

We present a connection between the Fisher dimension and graph structure via a curvature matrix. While the general relationship remains conjectural, we prove it exactly for chain graphs under weak-edge scaling.

**Definition 3.8** (Curvature Matrix). For a DAG $G$, define the curvature matrix $\mathcal{C}_G \in \mathbb{R}^{|E| \times |E|}$ with entries $(\mathcal{C}_G)_{e,e'} = \sum_{v \in V(G)} \mathbb{1}[e, e' \in \mathrm{MB}(v)]$, where $\mathrm{MB}(v)$ is the Markov blanket of $v$ and an edge $e$ is "in $\mathrm{MB}(v)$" if it lies in the

induced skeleton on $\{v\} \cup \mathrm{MB}(v)$. For the equivalence class: $\mathcal{C}_{[G]} = \sum_{G' \in [G]} \mathcal{C}_{G'}$.

**Conjecture 3.9** (Curvature Characterization). *For linear Gaussian SEMs with parameter scalings where $\mathcal{F}([G])$ can grow with $d$, there exist universal constants $c_1, c_2 > 0$ such that*
$$c_1 \cdot \lambda_{\min}(\mathcal{C}_{[G]})^{-1} \leq \mathcal{F}([G]) \leq c_2 \cdot \lambda_{\min}(\mathcal{C}_{[G]})^{-1},$$
*where $\lambda_{\min}$ denotes the minimum eigenvalue.*

*Remark* 3.10. This conjecture requires parameter regimes where $\mathcal{F}([G])$ grows with $d$. In the spectrally well-conditioned regime of Proposition 4.2, $\mathcal{F}([G]) = O(1)$ for all graphs regardless of the curvature matrix, so no purely structural quantity can characterize Fisher dimension universally.

We prove Conjecture 3.9 for chains under a specific weak-edge scaling:

**Theorem 3.11** (Curvature Characterization for Chains). *Let $G$ be the directed chain $X_1 \to X_2 \to \cdots \to X_d$ with unit noise variances $\sigma_i^2 = 1$ and weak-edge coefficients $\beta_{i,i+1} = 1/\sqrt{d}$. Then for all $d \geq 2$,*
$$4 \cdot \lambda_{\min}(\mathcal{C}_{[G]})^{-1} \leq \mathcal{F}([G]) \leq 2\pi^2 \cdot \lambda_{\min}(\mathcal{C}_{[G]})^{-1},$$
*with explicit values $\mathcal{F}([G]) = d + 1$ and $\lambda_{\min}(\mathcal{C}_{[G]}) = 4d \sin^2(\pi/(2d))$.*

*Proof Sketch.* For the chain MEC with $|[G]| = d$ orientations, the curvature matrix satisfies $\mathcal{C}_{[G]} = d(2I + A)$ where $A$ is the path adjacency matrix on $d - 1$ edges. The eigenvalues of $2I + A$ are $2 + 2\cos(k\pi/d)$ for $k = 1, \ldots, d-1$, giving $\lambda_{\min}(\mathcal{C}_{[G]}) = 4d \sin^2(\pi/(2d)) \in [4/d, \pi^2/d]$.

For the Fisher dimension, the canonical conditioning set for edge $(i, i+1)$ is empty, so $\rho_{\min} = \min_i |\mathrm{Corr}(X_i, X_{i+1})|$. With $\beta = 1/\sqrt{d}$, the minimum occurs at edge $(1, 2)$ where $\mathrm{Var}(X_1) = 1$ and $\mathrm{Var}(X_2) = 1 + 1/d$, giving $\rho_{\min} = 1/\sqrt{d+1}$ and $\mathcal{F}([G]) = d + 1$.

Combining: $\mathcal{F}([G]) \cdot \lambda_{\min}(\mathcal{C}_{[G]}) \in [4, 2\pi^2]$ for all $d \geq 2$. See Appendix G for the full proof. $\square$

## 4. Spectrally Well-Conditioned Regime

### 4.1. Partial Correlation Formula

For a linear Gaussian SEM, the partial correlations in the sufficient test set have a clean formula relating them to edge coefficients.

**Lemma 4.1** (Partial Correlation Formula). *For edge $(j, i) \in E(G)$ with coefficient $\beta_{ji}$, using the canonical conditioning set $S = Pa(i) \setminus \{j\}$,*
$$\rho_{ij|S} = \beta_{ji} \cdot \sqrt{\frac{Var(X_j|X_S)}{Var(X_i|X_S)}}.$$

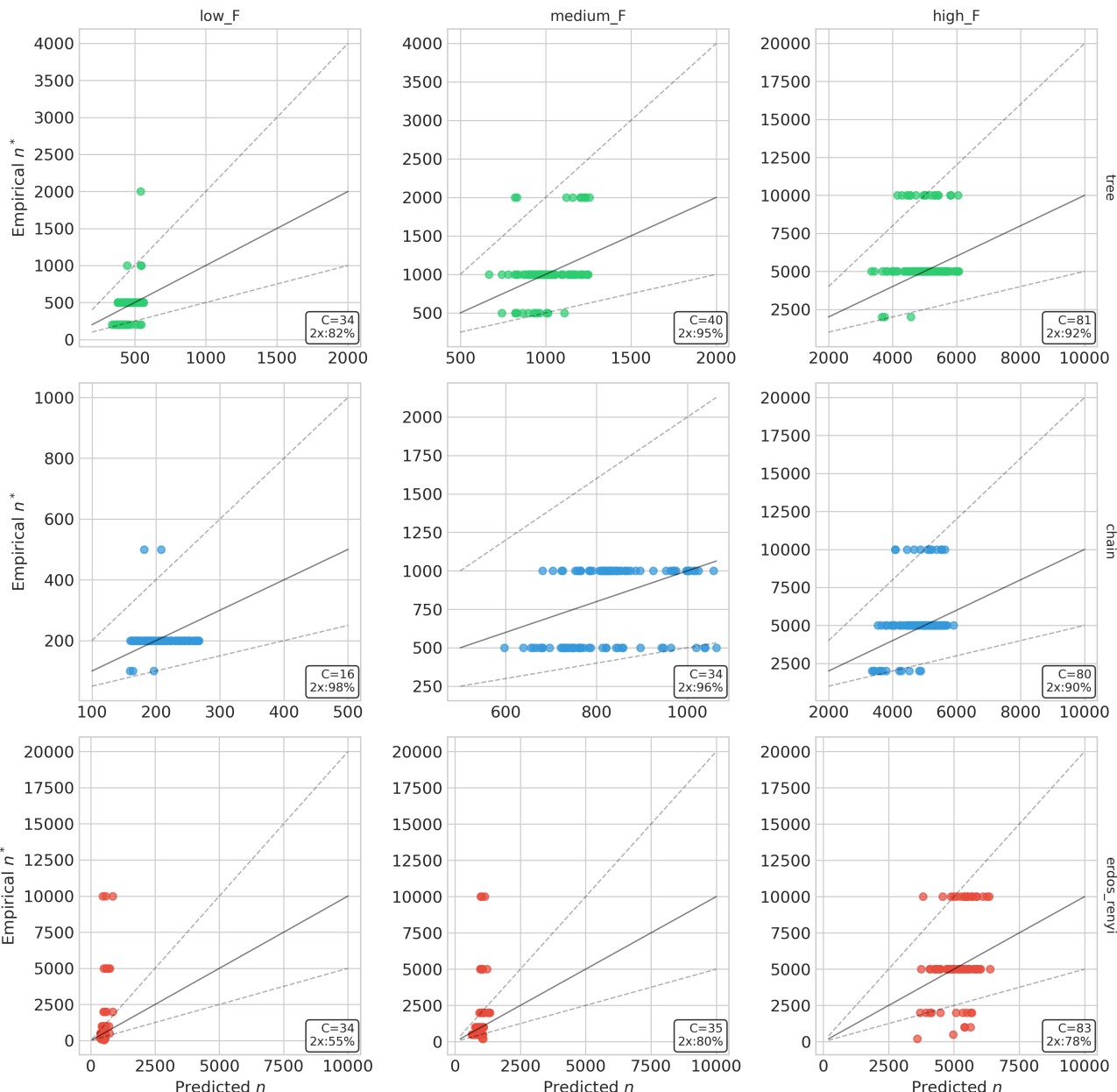

*Figure 1.* Predicted vs empirical sample complexity $n^*$ for three graph families and three edge coefficient regimes, with factor-of-2 bounds shown as dashed lines.

*Table 1.* Correlation between $\mathcal{F}([G]) \log(d/\delta)$ and empirical sample complexity $n^*$ for $d = 10$ graphs.

| GRAPH FAMILY | FITTED $C$ | PEARSON $r$ | SPEARMAN $\rho$ | $n$ GRAPHS |
|---|---|---|---|---|
| TREE | 40.2 | 0.87 | **0.91** | 500 |
| CHAIN | 32.2 | 0.92 | **0.94** | 500 |
| ERDŐS–RÉNYI | 42.1 | 0.55 | **0.66** | 500 |

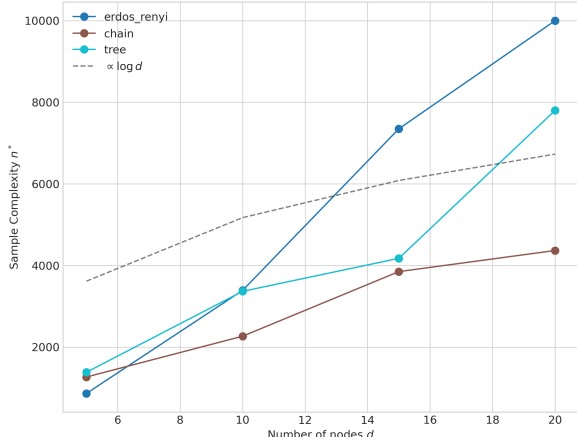

*Figure 2.* Sample complexity $n^*$ as a function of graph size $d$ for three graph families, with $\propto \log d$ reference line.

*Proof.* The proof follows from the SEM structure. Since $X_i = \beta_{ji} X_j + \sum_{k \in S} \beta_{ki} X_k + \varepsilon_i$ and $\varepsilon_i \perp\!\!\!\perp (X_j, X_S)$, we have $\mathrm{Cov}(X_i, X_j | X_S) = \beta_{ji} \cdot \mathrm{Var}(X_j | X_S)$. The partial correlation formula then gives the stated result. $\square$

### 4.2. Uniform Bound on Fisher Dimension

**Proposition 4.2.** *For a linear Gaussian SEM with minimum edge coefficient $|\beta_{ij}| \geq \varepsilon$ for all edges, noise variances bounded below by $\sigma_i^2 \geq \sigma_{\min}^2 > 0$ for all $i$, and covariance eigenvalues bounded above by $\lambda_{\max}(\Sigma) \leq C_\Sigma < \infty$, the Fisher dimension satisfies*

$$\mathcal{F}([G]) \leq \frac{C_\Sigma}{\varepsilon^2 \sigma_{\min}^2}.$$

In particular, for spectrally well-conditioned SEMs where $\sigma_{\min}^2, C_\Sigma = \Theta(1)$ and $\varepsilon = \Theta(1)$, we have $\mathcal{F}([G]) = O(1)$ independent of graph structure.

**Definition 4.3** (Spectrally Well-Conditioned SEM). A linear Gaussian SEM is spectrally well-conditioned if (i) noise variances are bounded away from zero: $\sigma_i^2 \geq \sigma_{\min}^2 = \Theta(1)$, (ii) covariance eigenvalues are bounded above: $\lambda_{\max}(\Sigma) \leq C_\Sigma = O(1)$, and (iii) edge coefficients satisfy $|\beta_{ij}| \geq \varepsilon = \Theta(1)$.

*Proof.* The proof bounds the variance ratio in Lemma 4.1. $\square$

Since $X_j = \sum_{k \in \mathrm{Pa}(j)} \beta_{kj} X_k + \varepsilon_j$ with $\varepsilon_j$ independent of $X_S$, we have $\mathrm{Var}(X_j | X_S) \geq \mathrm{Var}(\varepsilon_j) = \sigma_j^2 \geq \sigma_{\min}^2$. Conditioning can only reduce variance, so $\mathrm{Var}(X_i | X_S) \leq \mathrm{Var}(X_i) \leq \lambda_{\max}(\Sigma) \leq C_\Sigma$. Therefore $|\rho_{ij|S}| \geq \varepsilon \cdot \sigma_{\min}/\sqrt{C_\Sigma}$, giving $\rho_{\min} \geq \varepsilon \cdot \sigma_{\min}/\sqrt{C_\Sigma}$ and the stated bound on $\mathcal{F}([G])$. $\square$

This result has an important implication: for well-conditioned SEMs with $\Theta(1)$ parameters, the Fisher dimension does not distinguish between graph structures. Chains, trees, and complete DAGs all have $\mathcal{F}([G]) = O(1)$. Instance-specific variation in difficulty must therefore arise from parameter regimes where the well-conditioned assumptions are violated.

### 4.3. Regimes Where Fisher Dimension Grows

By Proposition 4.2, the Fisher dimension can grow with $d$ only if edge coefficients decay ($|\beta_{ij}| = O(1)$), noise variances shrink ($\sigma_{\min}^2 \to 0$), or covariance eigenvalues grow ($\lambda_{\max}(\Sigma) \to \infty$).

Consider a chain $X_1 \to X_2 \to \cdots \to X_d$ with $\beta_{i,i+1} = 1/\sqrt{d}$ and unit noise variances. For edge $(i, i+1)$, the canonical conditioning set is $S = \emptyset$. One can verify that $\mathrm{Var}(X_k) \in [1, 2)$ for all $k$, so $|\rho_{i,i+1}| = (1/\sqrt{d}) \cdot \Theta(1) = \Theta(1/\sqrt{d})$. Therefore $\rho_{\min} = \Theta(1/\sqrt{d})$ and $\mathcal{F}([G]) = \Theta(d)$.

In contrast, for a chain with unit coefficients ($\beta_{i,i+1} = 1$) and unit noise, although the maximum eigenvalue of the covariance matrix grows as $\lambda_{\max}(\Sigma) = \Theta(d^2)$, Lemma 4.1 gives $|\rho_{i,i+1}| = \sqrt{i/(i+1)} = \Theta(1)$. So with unit coefficients, $\rho_{\min} = \Theta(1)$ and $\mathcal{F}([G]) = \Theta(1)$ despite the growing eigenvalues.

### 4.4. Special Case: Complete DAGs

Unlike the conjectured scalings for chains and random graphs, the following result for complete DAGs is proven directly from Definition 3.1.

**Proposition 4.4** (Complete DAG Scaling). *For a complete DAG on $d$ nodes with edge coefficients $|\beta_{ij}| = \Theta(1)$, bounded noise variances, and bounded covariance eigenvalues*

$$\mathcal{F}([G]) = \Theta(1).$$

*Proof Sketch.* A complete DAG has $\binom{d}{2}$ edges and many v-structures, so $|[G]| = 1$ (the MEC is a singleton). For MEC recovery, PC must detect all adjacencies. With $\Theta(1)$ edge coefficients and spectrally well-conditioned parameters, Proposition 4.2 implies all relevant partial correlations are $\Theta(1)$. Therefore $\rho_{\min} = \Theta(1)$ and $\mathcal{F}([G]) = \Theta(1)$. See Appendix D for details. □

## 5. Computing the Fisher Dimension

### 5.1. Direct Computation from Partial Correlations

Since the Fisher dimension is defined as $\mathcal{F}([G]) = 1/\rho_{\min}^2$ with $\rho_{\min}$ taken over the sufficient test set (Definition 3.1), computation is straightforward given the DAG structure and covariance matrix.

---

**Algorithm 1** Compute Fisher Dimension

---

**Require:** DAG $G$, covariance matrix $\Sigma$
**Ensure:** Fisher dimension $\mathcal{F}([G])$
1: $\rho_{\min} \leftarrow \infty$
2: **for** each edge $(j, i) \in E(G)$ **do**
3: $\quad S \leftarrow \text{Pa}(i) \setminus \{j\}$ {Canonical conditioning set}
4: $\quad \rho \leftarrow \text{PartialCorr}(i, j, S; \Sigma)$
5: $\quad \rho_{\min} \leftarrow \min(\rho_{\min}, |\rho|)$
6: **end for**
7: **return** $1/\rho_{\min}^2$

---

The complexity of Algorithm 1 is $O(|E| \cdot d^3)$, as each partial correlation requires solving a linear system of size at most $d$.

*Remark* 5.1 (V-structures). V-structure detection requires PC to correctly identify that certain pairs $(i, j)$ are non-adjacent, i.e., that $\rho_{ij|S} = 0$ for some $S$. These *zero* partial correlations must be detected but do not enter the $\rho_{\min}$ computation (which is over *nonzero* correlations). The difficulty of v-structure detection is governed by the significance level, not by $\rho_{\min}$.

### 5.2. Concentration and Estimation from Data

In practice, the true DAG $G^*$ is unknown, so $\mathcal{F}([G^*])$ cannot be computed directly. However, we can estimate it from data and use it for post-hoc analysis.

**Lemma 5.2** (Concentration of Sample Partial Correlations). *Let $\hat{\rho}_{ij|S}$ be the sample partial correlation from $n$ i.i.d. samples. For $n > |S| + 3$, with probability $\geq 1 - \delta$,*

$$|\hat{\rho}_{ij|S} - \rho_{ij|S}| \leq C \cdot \sqrt{\frac{\log(1/\delta)}{n - |S| - 3}}$$

*for a universal constant $C > 0$.*

*Proof Sketch.* Fisher's z-transform of the sample partial correlation has approximately Gaussian distribution with vari-

ance $1/(n - |S| - 3)$. Applying standard Gaussian tail bounds yields the result. See Appendix E for details. □

**Proposition 5.3** (Convergence of Estimated Fisher Dimension). *For a fixed equivalence class $[G]$ with true minimum partial correlation $\rho_{\min} > 0$, with probability $\geq 1 - \delta$,*

$$\left|\hat{\mathcal{F}}_n([G]) - \mathcal{F}([G])\right| \leq O\left(\frac{1}{\rho_{\min}^3} \cdot \sqrt{\frac{\log(d/\delta)}{n}}\right),$$

*provided $n$ is large enough that $\hat{\rho}_{\min}$ is bounded away from zero.*

*Proof Sketch.* Apply Lemma 5.2 to each partial correlation in the sufficient test set with a union bound over $O(d^2)$ tests. The factor $1/\rho_{\min}^3$ arises from the delta method: for $f(\rho) = 1/\rho^2$, we have $|f'(\rho)| = 2/|\rho|^3$. See Appendix F for details. □

### 5.3. Practical Estimation

A *post-hoc analysis* approach is useful in practice:

1. Run a structure learning algorithm (e.g., PC) to obtain an estimated CPDAG $[\hat{G}]$

2. Compute $\mathcal{F}([\hat{G}])$ for the learned structure using the sample covariance matrix

3. Use $\mathcal{F}([\hat{G}])$ as a retrospective diagnostic for the difficulty of the learning task

This provides a *retrospective measure* of instance difficulty, useful for understanding why certain datasets required more samples, comparing relative difficulty across problems, and assessing confidence in learned structures (high $\mathcal{F}$ suggests the learning problem was hard, warranting caution). Because $\mathcal{F}([\hat{G}])$ depends on the learned structure, it should not be used to guide the same discovery process that produced $[\hat{G}]$.

This is not a sequential oracle or stopping rule. Lemma 5.2 and Proposition 5.3 give fixed-$n$ convergence guarantees for equivalence classes with $\rho_{\min} > 0$; they are not anytime-valid confidence sequences. At low sample sizes, $\hat{\mathcal{F}}$ can be unstable because $\mathcal{F} = 1/\rho_{\min}^2$ is singular as $\rho_{\min} \to 0$. Thus the diagnostic is most reliable after successful structure recovery and when the relevant nonzero partial correlations are bounded away from zero. Developing time-uniform confidence sequences for sequential stopping is future work, not a claim of the present analysis.

## 6. Empirical Validation

We provide empirical support for the Fisher dimension framework through four experiments. All experiments use the PC algorithm implementation from the

*Table 2.* Fraction of instances where $0.5 \leq n^*/n_{\text{pred}} \leq 2$.

| GRAPH FAMILY | LOW $\mathcal{F}$ | MEDIUM $\mathcal{F}$ | HIGH $\mathcal{F}$ |
|---|---|---|---|
| TREE | 82% | 95% | 92% |
| CHAIN | 98% | 96% | 90% |
| ERDŐS–RÉNYI | 55% | 80% | 78% |

`causal-learn` package, with Fisher's $z$-test for conditional independence. Noise variances are fixed at 1.0. We define the sample complexity $n^*$ as the smallest tested sample size for which the algorithm achieves exact recovery, measured by structural Hamming distance $= 0$, in at least 90% of 15 independent trials. The tested sample sizes range from 100 to 10,000 in multiplicative steps.

Experiments 6.1, 6.2, and 6.3 consider three graph types, trees, chains, and Erdős–Rényi graphs, with $d = 10$ nodes. We vary the edge-coefficient ranges to induce variation in Fisher dimension. Experiment 6.4 studies scaling with graph size for $d \in \{5, 10, 15, 20\}$. In Experiment 6.1, we generate 500 graphs of each type, Erdős–Rényi DAGs with edge probability 0.3, chains, and random spanning trees. In Experiments 6.2, 6.3, and 6.4, we use three coefficient families: low-$\mathcal{F}$, with coefficients in $(0.6, 0.9)$; medium-$\mathcal{F}$, with coefficients in $(0.4, 0.6)$; and high-$\mathcal{F}$, with coefficients in $(0.25, 0.4)$. The Fisher dimension is computed directly from the population covariance matrix using Definition 3.1.

### 6.1. Correlation Between Fisher Dimension and Sample Complexity

We generate 500 random DAGs per graph type with edge coefficients varied across five beta ranges from $(0.2, 0.4)$ to $(0.6, 0.9)$. For each graph, we compute both $\mathcal{F}([G]) \cdot \log(d/\delta)$ and the empirical sample complexity $n^*$ at which PC recovers the MEC with probability at least 0.9.

The result is summarized in Table 1. Tree and chain families show strong correlation between the theoretical predictor and empirical sample complexity (Spearman $\rho = 0.91$ and 0.94, respectively), while Erdős–Rényi graphs (Erdos & Renyi, 1960) show a positive but weaker correlation (Spearman $\rho = 0.66$). This pattern reflects the role of Fisher dimension as a measure of parametric detectability: within fixed or structured graph types, where skeleton and v-structure variation is limited, Fisher dimension captures the signal-strength variation that drives sample complexity. In more heterogeneous families such as Erdős–Rényi graphs, structural variation contributes additional difficulty, and structural proxies such as average Markov blanket size can be more predictive (Table 3).

### 6.2. Tightness of Upper and Lower Bounds

To examine how tight the theoretical scaling is in practice, we generate 100 graphs per cell in a 3 graph types $\times$ 3 beta families design (low $\mathcal{F}$, medium $\mathcal{F}$, high $\mathcal{F}$, corresponding to strong, medium, and weak edge coefficients). We fit a constant $C$ per cell in $n_{\text{pred}} = C \cdot \mathcal{F}([G]) \cdot \log d$ and compute the fraction of instances where the empirical $n^*$ falls within factor 2 of the prediction.

As shown in Table 2, for trees and chains, roughly 82-98% of instances have empirical sample complexity within factor 2 of the prediction, suggesting that the theoretical bounds are reasonably tight on these families. Erdős–Rényi graphs are noisier, with the low-$\mathcal{F}$ cell at 55%, but medium and high-$\mathcal{F}$ cells still achieve approximately 80%. Overall, 85% of predictions across all cells fall within factor 2. This agreement is further illustrated in Figure 1, which plots empirical sample complexity against the theoretical prediction and highlights the factor-of-2 tolerance bands.

### 6.3. Comparison with Alternative Complexity Proxies

We compare Fisher dimension to simpler structural proxies as predictors of sample complexity: graph density, maximum in-degree, average Markov blanket size, number of v-structures, MEC size estimate, a curvature-based proxy, and number of edges. The relative effectiveness of these proxies as predictors of empirical sample complexity is presented in Table 3.

Fisher dimension achieves the highest correlation for structured families, suggesting it captures parametric difficulty not explained by coarse structural statistics. Within a fixed graph type, structural proxies are approximately constant and do not vary between instances, whereas Fisher dimension varies with edge coefficient strength. For Erdős–Rényi graphs, where structural variance within the class dominates, structural proxies like Markov blanket size are more predictive.

### 6.4. Scaling with Graph Size

We vary $d \in \{5, 10, 15, 20\}$ for each family, generating 30 graphs per $(family, d)$ combination and estimating $n^*$ from 15 trials per sample size. The resulting empirical sample complexities across graph sizes are reported in Table 4.

In all three families, the empirical sample complexity increases with $d$ and correlates strongly with $\log d$ (Pearson $r > 0.92$). After normalizing by $\mathcal{F}([G]) \log d$, the coefficients of variation are below 0.5 for all families and substantially smaller for chains (0.18). Although the range of $d$ is limited, these patterns are consistent with the theoretical scaling $n^* \propto \mathcal{F}([G]) \log d$. The Erdős–Rényi family reaches a ceiling at $n^* = 10{,}000$ for $d = 20$, which may

*Table 3.* Spearman correlation between complexity proxies and empirical $n^*$.

| GRAPH FAMILY | BEST PROXY | BEST $r$ | FISHER DIM $r$ |
|---|---|---|---|
| TREE | FISHER DIMENSION | **0.88** | 0.88 |
| CHAIN | FISHER DIMENSION | **0.93** | 0.93 |
| ERDŐS–RÉNYI | AVG MARKOV BLANKET SIZE | 0.58 | 0.37 |

*Table 4.* Empirical sample complexity $n^*$ across graph sizes.

| FAMILY | $d = 5$ | $d = 10$ | $d = 15$ | $d = 20$ | CORR. WITH $\log d$ | CV |
|---|---|---|---|---|---|---|
| ERDŐS–RÉNYI | 863 | 3,390 | 7,350 | 10,000 | 0.97 | 0.43 |
| CHAIN | 1,267 | 2,267 | 3,850 | 4,367 | 0.98 | 0.18 |
| TREE | 1,383 | 3,367 | 4,175 | 7,800 | 0.92 | 0.36 |

compress the observed correlation. This scaling behavior is visualized in Figure 2, which illustrates the approximately logarithmic growth of sample complexity with graph size across graph families.

## 7. Discussion and Outlook

We introduced the Fisher dimension as an instance-dependent complexity measure for MEC recovery in linear Gaussian SEMs, establishing matching bounds (up to $\log d$). The definition captures minimum curvature among MEC-distinguishing directions in covariance space. Empirical validation confirms strong correlation between $\mathcal{F}([G]) \cdot \log d$ and observed sample complexity.

A central finding is Proposition 4.2: for spectrally well-conditioned SEMs, $\mathcal{F}([G]) = O(1)$ independent of graph structure. Instance-specific variation thus arises from parameter regimes with decaying edge strengths or growing covariance eigenvalues. We complement this with Proposition 4.4 ($\mathcal{F}([G]) = \Theta(1)$ for complete DAGs) and Theorem 3.11, which proves $\mathcal{F}([G]) \asymp \lambda_{\min}(\mathcal{C}_{[G]})^{-1}$ for chains under weak-edge scaling. Extending this curvature characterization to general graphs remains open.

For practical use, Algorithm 1 computes the Fisher dimension efficiently, and post-hoc estimation (Section 5.3) provides convergence guarantees. Limitations include restriction to linear Gaussian SEMs and the faithfulness assumption. A concrete open direction is to combine Fisher dimension with a graph-dependent effective search complexity, replacing the generic union-bound factor $O(\log d)$ by a term such as $N_{\text{eff}}([G])$ that better reflects the structural search complexity of the MEC. Other future directions include adaptive algorithms, extensions to interventional data (He & Geng, 2008) and latent confounders, and non-Gaussian settings (Wang & Drton, 2020; 2023).

## Impact Statement

This paper presents work whose goal is to advance the field of machine learning by providing theoretical foundations for understanding instance-specific difficulty in causal discovery. The results are primarily theoretical and methodological, with potential applications to improving the efficiency and reliability of causal inference in scientific domains. We do not anticipate specific negative societal consequences that require highlighting, though as with any work that improves causal inference capabilities, downstream applications should be evaluated for their own ethical implications.

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

## A. Additional Definitions

**Definition A.1** (Markov Blanket). For a DAG $G = (V, E)$ and node $v \in V$, the Markov blanket of $v$ is

$$\mathrm{MB}_G(v) = \mathrm{Pa}_G(v) \cup \mathrm{Ch}_G(v) \cup \Big( \bigcup_{u \in \mathrm{Ch}_G(v)} \mathrm{Pa}_G(u) \Big) \setminus \{v\},$$

where $\mathrm{Pa}_G(v)$ and $\mathrm{Ch}_G(v)$ denote the parents and children of $v$, respectively. Equivalently, $\mathrm{MB}_G(v)$ consists of the parents of $v$, the children of $v$, and the other parents of those children. When the graph is clear, we write $\mathrm{MB}(v)$ or $\mathrm{MB}(v)$. In a distribution Markov with respect to $G$, $X_v$ is conditionally independent of all variables outside $\{v\} \cup \mathrm{MB}_G(v)$ given $X_{\mathrm{MB}_G(v)}$.

**Definition A.2** (Fisher–Rao Metric). For a smooth parametric family $\{p_\theta : \theta \in \Theta\}$, the Fisher–Rao metric is the Riemannian metric whose coordinate representation is the Fisher information matrix

$$g_\theta(a, b) = a^\top I(\theta) b, \qquad I(\theta)_{ij} = \mathbb{E}_\theta \big[ \partial_{\theta_i} \log p_\theta(X) \, \partial_{\theta_j} \log p_\theta(X) \big],$$

for tangent directions $a, b \in T_\theta \Theta$. For zero-mean Gaussian distributions parameterized by the covariance matrix $\Sigma$, tangent directions are symmetric matrices $A, B$, and the metric takes the form

$$g_\Sigma(A, B) = \frac{1}{2} \operatorname{tr}\big(\Sigma^{-1} A \Sigma^{-1} B\big).$$

This is the formula used in Remark 3.2 for the one-dimensional correlation submanifold $\Sigma(\rho)$.

## B. Full Proof of Upper Bound on Sample Complexity

*Proof of Theorem 3.4.*

### Step 1: PC Algorithm and Conditional Independence Tests

The PC algorithm recovers $[G^*]$ by testing conditional independencies. For each pair $(i, j)$ and conditioning set $S$ with $|S| \leq \Delta$, it tests (i) $H_0$: $\rho_{ij|S} = 0$ (conditional independence) and (ii) $H_1$: $\rho_{ij|S} \neq 0$ (conditional dependence).

The total number of tests is at most:

$$T = \binom{d}{2} \cdot \sum_{k=0}^{\Delta} \binom{d-2}{k} \leq d^2 \cdot (d-2)^\Delta / \Delta! = O(d^{2+\Delta})$$

### Step 2: Partial Correlation Estimation

Given $n$ i.i.d. samples from a multivariate Gaussian, the sample partial correlation $\hat{\rho}_{ij|S}$ can be computed from the sample covariance matrix. Fisher's z-transform:

$$\hat{z}_{ij|S} = \frac{1}{2} \log \frac{1 + \hat{\rho}_{ij|S}}{1 - \hat{\rho}_{ij|S}}$$

satisfies, for $n > |S| + 3$:

$$\hat{z}_{ij|S} \sim \mathcal{N}\left( z^*_{ij|S}, \frac{1}{n - |S| - 3} \right)$$

where $z^*_{ij|S} = \frac{1}{2} \log \frac{1+\rho^*_{ij|S}}{1-\rho^*_{ij|S}}$ is the population z-value. For $|\rho| < 0.5$, we have $|z| \approx |\rho|$ up to a factor of 1.1.

### Step 3: Hypothesis Test and Error Probability

Use the two-sided z-test at level $\alpha$: reject $H_0$ if $|\hat{z}_{ij|S}| \cdot \sqrt{n - |S| - 3} > z_{\alpha/2}$.

**Type I error:** Under $H_0$ ($\rho_{ij|S} = 0$), $\mathbb{P}(\text{reject } H_0) = \alpha$.

**Type II error:** Under $H_1$ with $|\rho_{ij|S}| \geq \rho_{\min}$, the power is:

$$\text{Power} = \mathbb{P}(\text{reject } H_0 \mid H_1) \geq 1 - \Phi\left(z_{\alpha/2} - |z^*|\sqrt{n - |S| - 3}\right)$$

For $|z^*| \geq 0.9\rho_{\min}$ (valid when $\rho_{\min} < 0.5$), to achieve power $\geq 1 - \beta$:

$$\sqrt{n - |S| - 3} \geq \frac{z_{\alpha/2} + z_\beta}{0.9\rho_{\min}}$$

Squaring and rearranging:

$$n \geq |S| + 3 + \frac{(z_{\alpha/2} + z_\beta)^2}{0.81\rho_{\min}^2}$$

**Step 4: Union Bound**

To ensure all $T$ tests succeed with probability $\geq 1 - \delta$, apply Bonferroni: set $\alpha = \beta = \delta/(2T)$.

Then $z_{\alpha/2} = z_{\delta/(4T)} \leq \sqrt{2\log(4T/\delta)}$ and similarly for $z_\beta$.

Substituting:

$$(z_{\alpha/2} + z_\beta)^2 \leq 8\log(4T/\delta) \leq 8\log(4d^{2+\Delta}/\delta) = O\left((2 + \Delta)\log d + \log(1/\delta)\right)$$

For bounded $\Delta$, this is $O(\log(d/\delta))$.

**Step 5: Final Sample Complexity**

Combining the above:

$$n \geq \Delta + 3 + \frac{C_1 \log(d/\delta)}{\rho_{\min}^2}$$

For $n$ large enough, the $\Delta + 3$ term is absorbed, giving:

$$n \geq C \cdot \frac{\log(d/\delta)}{\rho_{\min}^2} = C \cdot \mathcal{F}([G^*]) \cdot \log(d/\delta)$$

where $C$ depends on $\Delta$ through the number of tests. $\qquad\square$

## C. Full Proof of Lower Bound on Sample Complexity

*Proof of Theorem 3.6.* We use Le Cam's two-point method.

**Step 1: Setup**

Let $P_0 = \mathcal{N}(0, \Sigma_0)$ and $P_1 = \mathcal{N}(0, \Sigma_1)$ be the Gaussian distributions corresponding to $G_0$ and $G_1$. By assumption, they differ only in the $(i, j)$ conditional relationship given $X_S$.

**Step 2: KL Divergence Computation**

For multivariate Gaussians:

$$\text{KL}(P_0\|P_1) = \frac{1}{2}\left[\text{tr}(\Sigma_1^{-1}\Sigma_0) - d + \log\frac{|\Sigma_1|}{|\Sigma_0|}\right]$$

When $\Sigma_0$ and $\Sigma_1$ differ only in the $(i, j|S)$ conditional correlation, we can use the chain rule for KL divergence. Since the marginal on $X_S$ is identical under both:

$$\text{KL}(P_0\|P_1) = \mathbb{E}_{X_S}[\text{KL}(P_0(X_i, X_j|X_S)\|P_1(X_i, X_j|X_S))]$$

For bivariate Gaussians with unit conditional variances and correlation $\rho$ vs 0, the KL divergence has closed form:

$$\text{KL}(\mathcal{N}_\rho\|\mathcal{N}_0) = -\frac{1}{2}\log(1 - \rho^2)$$

Taylor expanding at $\rho = 0$:

$$-\frac{1}{2}\log(1 - \rho^2) = \frac{\rho^2}{2} + \frac{\rho^4}{4} + O(\rho^6)$$

More generally, for bivariate Gaussians with arbitrary (but equal) conditional variances, the leading term remains $O(\rho^2)$; only the constant changes. Therefore, for small $\rho_0$:

$$\mathrm{KL}(P_0 \| P_1) \leq C\rho_0^2$$

for some constant $C$ depending on the conditional variances. For simplicity, we absorb this into the universal constant $c$ in the theorem statement.

For $n$ i.i.d. samples:

$$\mathrm{KL}(P_0^n \| P_1^n) = n \cdot \mathrm{KL}(P_0 \| P_1) \leq n\rho_0^2$$

**Step 3: Le Cam's Lemma**

Le Cam's lemma states that for any test $\psi : \mathcal{X}^n \to \{0, 1\}$:

$$\mathbb{P}_{P_0^n}(\psi = 1) + \mathbb{P}_{P_1^n}(\psi = 0) \geq 1 - \mathrm{TV}(P_0^n, P_1^n)$$

By Pinsker's inequality:

$$\mathrm{TV}(P_0^n, P_1^n) \leq \sqrt{\frac{1}{2}\mathrm{KL}(P_0^n \| P_1^n)} \leq \sqrt{\frac{n\rho_0^2}{2}}$$

**Step 4: Conclude**

For the sum of error probabilities to be at least $1/2$:

$$1 - \sqrt{\frac{n\rho_0^2}{2}} \geq \frac{1}{2}$$

which requires $n \leq 2/\rho_0^2$. Setting $c = 2$ completes the proof. $\square$

*Proof of Corollary 3.7.* By Definition 3.1, $\rho_{\min}$ is the smallest partial correlation in the sufficient test set $\mathcal{T}([G])$. For each such test $(i, j, S)$ with $\rho_{ij|S} = \rho_0 \neq 0$, there exists an alternative MEC $[G']$ where edge $(i, j)$ is absent (so $\rho_{ij|S} = 0$ under $G'$). Theorem 3.6 gives $n = \Omega(1/\rho_0^2)$. Taking $\rho_0 = \rho_{\min}$:

$$n = \Omega\left(\frac{1}{\rho_{\min}^2}\right) = \Omega(\mathcal{F}([G]))$$

$\square$

# D. Proof of Complete DAG Scaling

*Proof of Proposition 4.4.* Consider a complete DAG on $d$ nodes with some topological ordering $\pi$. Every pair of nodes is connected, so there are $\binom{d}{2}$ edges. Every triple $i < j < k$ (in the topological order) forms a v-structure $i \to k \leftarrow j$ since all pairs are adjacent except we must check: actually in a complete DAG, $i \to j$ and $j \to k$ and $i \to k$ are all present, so there are no v-structures in the usual sense.

More precisely, in a complete DAG, every unshielded triple $i - k - j$ has $i$ and $j$ adjacent (since the graph is complete), so there are no unshielded triples and hence no v-structures. However, the MEC is still a singleton because the complete graph has a unique CPDAG representation (all edges directed according to the unique topological ordering that makes the graph acyclic).

For MEC recovery, we need to detect all $\binom{d}{2}$ adjacencies. For each edge $(j, i)$ with $j \to i$ in the DAG, the canonical conditioning set is $S = \mathrm{Pa}(i) \setminus \{j\}$. By Lemma 4.1:

$$|\rho_{ij|S}| = |\beta_{ji}| \cdot \sqrt{\frac{\mathrm{Var}(X_j|X_S)}{\mathrm{Var}(X_i|X_S)}}$$

Under the well-conditioned assumptions ($|\beta_{ji}| \geq \varepsilon = \Theta(1)$, $\sigma_i^2 \geq \sigma_{\min}^2 = \Theta(1)$, and $\lambda_{\max}(\Sigma) \leq C_\Sigma = O(1)$), Proposition 4.2 gives:

$$|\rho_{ij|S}| \geq \frac{\varepsilon \sigma_{\min}}{\sqrt{C_\Sigma}} = \Theta(1)$$

Since this bound holds for all edges, $\rho_{\min} = \Theta(1)$ and therefore $\mathcal{F}([G]) = 1/\rho_{\min}^2 = \Theta(1)$. $\qquad\square$

## E. Proof of Concentration Lemma

*Proof of Lemma 5.2.* Fisher's z-transform of the sample partial correlation is defined as:

$$\hat{z}_{ij|S} = \frac{1}{2} \log \frac{1 + \hat{\rho}_{ij|S}}{1 - \hat{\rho}_{ij|S}}$$

For $n$ i.i.d. samples from a multivariate Gaussian with $n > |S| + 3$, the distribution of $\hat{z}_{ij|S}$ is approximately:

$$\hat{z}_{ij|S} \sim \mathcal{N}\left(z_{ij|S}^*, \frac{1}{n - |S| - 3}\right)$$

where $z_{ij|S}^* = \frac{1}{2} \log \frac{1 + \rho_{ij|S}}{1 - \rho_{ij|S}}$ is the population z-value.

By standard Gaussian tail bounds, for any $t > 0$:

$$\mathbb{P}\left(|\hat{z}_{ij|S} - z_{ij|S}^*| > t\right) \leq 2 \exp\left(-\frac{t^2(n - |S| - 3)}{2}\right)$$

Setting the right-hand side equal to $\delta$ and solving for $t$:

$$t = \sqrt{\frac{2 \log(2/\delta)}{n - |S| - 3}}$$

For small partial correlations (say $|\rho| < 0.5$), the z-transform satisfies $|z - \rho| = O(\rho^3)$, so $|z| \approx |\rho|$. More precisely, $\rho = \tanh(z)$ implies:

$$\left|\frac{\partial \rho}{\partial z}\right| = 1 - \tanh^2(z) = 1 - \rho^2 \in [0.75, 1]$$

for $|\rho| < 0.5$. Therefore, concentration of $\hat{z}$ implies concentration of $\hat{\rho}$:

$$|\hat{\rho}_{ij|S} - \rho_{ij|S}| \leq \frac{|\hat{z}_{ij|S} - z_{ij|S}^*|}{1 - \rho_{\max}^2} \leq C \cdot \sqrt{\frac{\log(1/\delta)}{n - |S| - 3}}$$

where $C$ is a universal constant (we can take $C = 2$ for $|\rho| < 0.5$). $\qquad\square$

# F. Proof of Fisher Dimension Convergence

*Proof of Proposition 5.3.* Let $\mathcal{T}([G])$ be the sufficient test set from Definition 3.1, with $|\mathcal{T}([G])| \leq |E(G)| \leq d^2$ tests. By Lemma 5.2, for each test $(i, j, S) \in \mathcal{T}([G])$:

$$|\hat{\rho}_{ij|S} - \rho_{ij|S}| \leq C \cdot \sqrt{\frac{\log(d^2/\delta')}{n - |S| - 3}}$$

with probability $\geq 1 - \delta'$.

Applying a union bound over all tests with $\delta' = \delta/d^2$:

$$\max_{(i,j,S) \in \mathcal{T}([G])} |\hat{\rho}_{ij|S} - \rho_{ij|S}| \leq C \cdot \sqrt{\frac{\log(d^4/\delta)}{n - \Delta - 3}} = O\left(\sqrt{\frac{\log(d/\delta)}{n}}\right)$$

with probability $\geq 1 - \delta$, where $\Delta$ is the maximum conditioning set size.

Let $\hat{\rho}_{\min} = \min_{(i,j,S) \in \mathcal{T}([G]), \rho_{ij|S} \neq 0} |\hat{\rho}_{ij|S}|$ be the estimated minimum nonzero partial correlation. Then:

$$|\hat{\rho}_{\min} - \rho_{\min}| \leq \max_{(i,j,S)} |\hat{\rho}_{ij|S} - \rho_{ij|S}| = O\left(\sqrt{\frac{\log(d/\delta)}{n}}\right)$$

For the Fisher dimension $\mathcal{F}([G]) = 1/\rho_{\min}^2$, we use the delta method. Let $f(\rho) = 1/\rho^2$. Then:

$$|f'(\rho)| = \frac{2}{|\rho|^3}$$

By Taylor expansion around $\rho_{\min}$:

$$\left|\hat{\mathcal{F}}_n([G]) - \mathcal{F}([G])\right| = \left|\frac{1}{\hat{\rho}_{\min}^2} - \frac{1}{\rho_{\min}^2}\right| \leq \frac{2}{\rho_{\min}^3} \cdot |\hat{\rho}_{\min} - \rho_{\min}| + O(|\hat{\rho}_{\min} - \rho_{\min}|^2)$$

Provided $n$ is large enough that $|\hat{\rho}_{\min} - \rho_{\min}| < \rho_{\min}/2$ (ensuring $\hat{\rho}_{\min} > \rho_{\min}/2$), the higher-order terms are negligible, giving:

$$\left|\hat{\mathcal{F}}_n([G]) - \mathcal{F}([G])\right| \leq O\left(\frac{1}{\rho_{\min}^3} \cdot \sqrt{\frac{\log(d/\delta)}{n}}\right)$$

$\square$

# G. Full Proof of Curvature Characterization for Chains

*Proof of Theorem 3.11.* **Step 1: Compute $\lambda_{\min}(\mathcal{C}_{[G]})$ for the chain MEC.**

Let the skeleton edges be $e_k = \{k, k+1\}$ for $k = 1, \ldots, d-1$. For any orientation $G' \in [G]$ of the chain (all $d$ orientations are v-structure-free), each node $v$ has Markov blanket equal to its neighbors in the path. Thus $\{v\} \cup \text{MB}(v)$ contains the edges incident to $v$.

For a fixed orientation $G'$, the curvature matrix $\mathcal{C}_{G'}$ is the Gram matrix of vertex-edge incidence vectors on the path:

$$\mathcal{C}_{G'} = 2I_{d-1} + A$$

where $A$ is the adjacency matrix of the path graph on $d-1$ edge-nodes (i.e., $A_{k,k+1} = A_{k+1,k} = 1$ for $k = 1, \ldots, d-2$).

All orientations give the same $\mathcal{C}_{G'}$, and $|[G]| = d$, so:

$$\mathcal{C}_{[G]} = \sum_{G' \in [G]} \mathcal{C}_{G'} = d(2I + A)$$

The eigenvalues of the tridiagonal Toeplitz matrix $2I + A$ are:

$$\lambda_k(2I + A) = 2 + 2\cos\left(\frac{k\pi}{d}\right), \quad k = 1, 2, \ldots, d-1$$

The minimum eigenvalue is attained at $k = d - 1$:

$$\lambda_{\min}(2I + A) = 2 + 2\cos\left(\frac{(d-1)\pi}{d}\right) = 2 - 2\cos\left(\frac{\pi}{d}\right) = 4\sin^2\left(\frac{\pi}{2d}\right)$$

Multiplying by $d$:

$$\lambda_{\min}(\mathcal{C}_{[G]}) = 4d\sin^2\left(\frac{\pi}{2d}\right)$$

Using the bounds $\frac{2}{\pi}x \le \sin x \le x$ for $x \in [0, \pi/2]$ with $x = \frac{\pi}{2d}$:

$$\sin^2\left(\frac{\pi}{2d}\right) \in \left[\frac{1}{d^2}, \frac{\pi^2}{4d^2}\right]$$

Therefore:

$$\frac{4}{d} \le \lambda_{\min}(\mathcal{C}_{[G]}) \le \frac{\pi^2}{d}$$

Equivalently:

$$\frac{d}{\pi^2} \le \lambda_{\min}(\mathcal{C}_{[G]})^{-1} \le \frac{d}{4}$$

**Step 2: Compute $\mathcal{F}([G])$ for the chain SEM with $\beta = 1/\sqrt{d}$.**

In the chain $X_1 \to X_2 \to \cdots \to X_d$, the canonical conditioning set for each edge $(i, i+1)$ is:

$$S = \text{Pa}(i+1) \setminus \{i\} = \varnothing$$

So the relevant partial correlations are the adjacent correlations $\rho_{i,i+1|\varnothing} = \text{Corr}(X_i, X_{i+1})$.

With the SEM $X_1 = \varepsilon_1$ and $X_{i+1} = \beta X_i + \varepsilon_{i+1}$ where $\varepsilon_i \sim \mathcal{N}(0, 1)$:

$$\text{Cov}(X_i, X_{i+1}) = \beta \cdot \text{Var}(X_i), \quad \text{Var}(X_{i+1}) = \beta^2 \text{Var}(X_i) + 1$$

Hence:

$$|\rho_{i,i+1}| = \frac{|\beta|\text{Var}(X_i)}{\sqrt{\text{Var}(X_i)\text{Var}(X_{i+1})}} = |\beta|\sqrt{\frac{\text{Var}(X_i)}{1 + \beta^2 \text{Var}(X_i)}}$$

The function $f(V) = V/(1 + \beta^2 V)$ is increasing in $V > 0$, and $\text{Var}(X_i)$ increases with $i$, so the minimum adjacent correlation occurs at the first edge $(1, 2)$ where $\text{Var}(X_1) = 1$ and $\text{Var}(X_2) = 1 + \beta^2$:

$$\rho_{\min}([G]) = |\rho_{1,2}| = \frac{|\beta|}{\sqrt{1 + \beta^2}}$$

Setting $\beta = 1/\sqrt{d}$:

$$\rho_{\min}([G]) = \frac{1/\sqrt{d}}{\sqrt{1 + 1/d}} = \frac{1}{\sqrt{d+1}}$$

Therefore:

$$\mathcal{F}([G]) = \frac{1}{\rho_{\min}^2} = d + 1$$

**Step 3: Compare $\mathcal{F}([G])$ to $\lambda_{\min}(\mathcal{C}_{[G]})^{-1}$.**

From Step 1: $\lambda_{\min}(\mathcal{C}_{[G]})^{-1} \leq d/4$.

So: $\mathcal{F}([G]) = d + 1 \geq d \geq 4 \cdot \lambda_{\min}(\mathcal{C}_{[G]})^{-1}$.

Also from Step 1: $\lambda_{\min}(\mathcal{C}_{[G]})^{-1} \geq d/\pi^2$.

For $d \geq 2$, we have $d + 1 \leq 2d$, hence:

$$\mathcal{F}([G]) = d + 1 \leq 2d \leq 2\pi^2 \cdot \lambda_{\min}(\mathcal{C}_{[G]})^{-1}$$

Combining both bounds, for all $d \geq 2$:

$$4 \cdot \lambda_{\min}(\mathcal{C}_{[G]})^{-1} \leq \mathcal{F}([G]) \leq 2\pi^2 \cdot \lambda_{\min}(\mathcal{C}_{[G]})^{-1}$$

This establishes $\mathcal{F}([G]) \asymp \lambda_{\min}(\mathcal{C}_{[G]})^{-1}$ with explicit constants $c_1 = 4$ and $c_2 = 2\pi^2 \approx 19.74$.

**Remark on asymptotics.** As $d \to \infty$, $\sin(\pi/(2d)) \sim \pi/(2d)$, so $\lambda_{\min}(\mathcal{C}_{[G]}) \sim \pi^2/d$ and $\mathcal{F}([G]) \cdot \lambda_{\min}(\mathcal{C}_{[G]}) \to \pi^2 \approx 9.87$. $\qquad\square$

**Extension to other graphs.** Similar analysis can be applied to star graphs and other families. The key requirement is a parameter scaling (e.g., $\beta = 1/\sqrt{d}$) that causes $\mathcal{F}([G])$ to grow with $d$; under well-conditioned parameters, $\mathcal{F}([G]) = O(1)$ regardless of structure, and the curvature characterization does not apply.

