# OpenReview forum: "The Fisher Dimension: Instance-Dependent Complexity for Causal Discovery"
_ICML.cc/2026/Conference — ICML 2026 regular_

### Official Review · Reviewer_kQgc · 2026-03-11

**Soundness:** 3
**Presentation:** 3
**Significance:** 3
**Originality:** 3
**Overall Recommendation:** 4
**Confidence:** 3

**Summary:**

The paper studies the instance-specific complexity for Markov equivalence
class (MEC) recovery in linear Gaussian
structural equation models. Specifically, the authors introduce the
Fisher dimension, defined as the inverse squared
minimum partial correlation that must be detected
to recover the MEC, and show that the Fisher
dimension provides both a lower bound and an upper bound
 for MEC recovery. They also show that the Fisher dimension
is uniformly bounded regardless of graph
structure, under certain conditions.

**Compliance With Llm Reviewing Policy:**

Affirmed.

**Final Justification:**

Most of my concerns are addressed and I keep the rating unchanged.

**Key Questions For Authors:**

1. What is the intuition that for complex graphs, e.g., ER families, the correlation between fisher dimension and empirical n* is lower?

2. How do you envision a practitioner using the Fisher Dimension for a priori sample size planning when the true structure is unknown? (especially given that the true structure is often complex and beyond the tree and chain family)

3. Is it possible to have experiments on even larger graphs (e.g., d=100)?

4. In experiments the empirical sample complexity is defined as n* at which
PC recovers the MEC with probability at least 0.9. Why 0.9 is chosen?

5. What if we define the fisher dimension by minimal $\rho$ over all the conditioning set rather than the canonical sufficient conditioning set?

**Limitations:**

It would help to have a discussion regarding the circularity of using structure found by a causal discovery method to predict the sample complexity of causal discovery methods.

**Strengths And Weaknesses:**

Strengths:

Theoretical Soundness: The paper is theoretically rigorous, providing near-tight upper and lower bounds on sample complexity for MEC recovery.

Presentation: The paper is well-written and each important theorem follows a proof sketch.

Algorithm Independence: The definition of the Fisher Dimension is algorithm-independent (but depends on the true structure which is unknown, see weaknesses below).

Weaknesses:

Practical Utility and Circularity: A major concern is the practical application of the proposed Fisher dimension. Calculating $\mathcal{F}([G])$ requires knowledge of the underlying causal structure or its MEC. In practice, if a researcher needs to know the complexity of discovering a graph, they typically do not have the graph yet. The suggestion to estimate the structure by a causal discovery method first introduces a circular dependency: the estimation process itself is subject to the statistical errors that the Fisher Dimension is meant to quantify.

Performance for complex graphs (e.g. Erdos-Renyi Family): Empirical results for the ER graph family show a significantly weaker correlation between the Fisher Dimension and empirical sample complexity compared to that of simpler graphs (tree or chain). Further, it would also be interesting to see the performance on even larger graphs (e.g., d=100).

---

> ### Author Rebuttal · Authors · 2026-03-30
>
> We sincerely thank the reviewer for their thoughtful feedback and for recognizing the theoretical rigor and clarity of our work.
>
> ## Weaknesses
>
> - We appreciate the reviewer’s point regarding the circularity of using an estimated structure to calculate the Fisher dimension (FD); this is a fair observation that we intend to clarify in the final version. About Practical Utility, as outlined in Section 5.3, we view the FD primarily as a diagnostic and retrospective tool rather than a pure a priori planning instrument. By computing $\hat{\mathcal{F}}$ on a learned CPDAG, researchers can gain a retrospective understanding of why a specific dataset was challenging to orient, compare relative difficulty across different problems, and flag potentially fragile learned structures.
>
> - The reviewer correctly notes that the correlation for Erdos-Renyi (ER) graphs is lower, and we appreciate this insight into the limitations of the current framework. Our results suggest that the FD is most effective when structural heterogeneity is controlled (as in trees or chains), allowing it to capture parametric variation in edge strengths. In ER families, structural variance across instances, such as significant differences in Markov blanket sizes, often dominates the difficulty, which explains why structural proxies sometimes outperform the FD in that specific setting.
>
> ## Answering Questions
>
> **1. What is the intuition that for complex graphs, e.g., ER families, the correlation between FD and empirical $n^*$ is lower?**
>
> The FD specifically tracks the weakest partial correlation required for recovery, essentially measuring parametric difficulty. In structured families like chains, the graph architecture is nearly constant, so the variation in sample complexity is driven almost entirely by the edge coefficients captured by $\mathcal{F}([G])$. In contrast, ER graphs have massive structural variation between instances; since structural variance dominates the difficulty in these cases, structural proxies like the average Markov blanket size become more predictive than a purely correlation-based measure.
>
> **2. How do you envision a practitioner using the FD for a priori sample size planning when the true structure is unknown?**
>
> While the paper does not justify pure a priori use without any data, it supports a sequential interpretation. A practitioner could run a first-pass causal discovery on a subset of data, compute $\hat{\mathcal{F}}$ on the resulting CPDAG, and use that value as a diagnostic to determine if more samples are needed to gain confidence in the orientations. This frames the FD as a tool for assessing the hardness of a specific instance after an initial structure is proposed.
>
> **3. Is it possible to have experiments on even larger graphs (e.g., d=100)?**
>
> We agree that $d=100$ would provide further validation. However, due to the computational intensity of estimating $n^*$, which requires many trials across various sample sizes, we capped our current experiments at $d=20$ to avoid excessive censoring at our $10$K sample limit.
>
> **4. In experiments the empirical sample complexity is defined as $n^{*}$ at which PC recovers the MEC with probability at least 0.9. Why 0.9 is chosen?**
>
> The $0.9$ threshold is an operational choice rather than a mathematical requirement. According to our theoretical scaling $n \propto \mathcal{F}([G]) \log(d/\delta)$, changing the required success probability from $0.9$ to $0.99$ would only rescale $n^*$ by a constant factor (approximately $1.5\times$ at $d=10$) rather than changing the underlying qualitative relationship between the FD and sample complexity.
>
> **5. What if we define the FD by minimal $|\rho_{ij|S}|$ over all the conditioning set rather than the canonical sufficient conditioning set?**
>
> Defining $\mathcal{F}$ over all possible conditioning sets would result in a much more pessimistic and potentially misleading measure. The canonical conditioning set $S = \text{Pa}(i) \setminus j$ represents the specific signal required by algorithms like PC to distinguish adjacencies. Minimizing over all $S$ would include irrelevant or unnecessarily difficult tests (such as conditioning on descendants), which does not accurately reflect the information-theoretic requirements for MEC recovery.

---

> > ### Author Rebuttal · Reviewer_kQgc · 2026-04-03
> >
> > Thank you for the response. Most of my concerns are addressed and I keep my rating unchanged.

---

### Official Review · Reviewer_yytz · 2026-03-11

**Soundness:** 3
**Presentation:** 3
**Significance:** 2
**Originality:** 3
**Overall Recommendation:** 5
**Confidence:** 2

**Summary:**

The authors introduce upper and lower bounds (coinciding modulo a log factor) for the sample complexity of learning a Markov equivalence class from linear-Gaussian data. These bounds depend on the Fisher dimension of the true MEC (or graph), a novel measure introduced by the authors. They show conditions under which the Fisher dimension does not depend on the graph structure or size but the parameters of the linear-Gaussian SEM, and conditions under which it does depend on the graph size. Experiments show that Fisher dimension is a good measure of empirical sample complexity when the true graph is a chain or a tree.

**Compliance With Llm Reviewing Policy:**

Affirmed.

**Final Justification:**

The rebuttal period clarified how the informativeness of the authors' bounds compares with that of Gao et al 2022. However, the authors' bounds can not be computed a priori, and it remains open, after the discussion period, whether the correctness of the learned MEC is necessary for the post-hoc bounds to be informative. My expectation was that this would be clarified in more depth following my review, absent which I must assume the answer to the latter question is yes. **I maintain my initial score with an added caveat.** While the authors' work provides a new insight into how the hardness of learning depends on the parameters (and not just structure) of the underlying SEM, I am less convinced that their bounds are useful in practice.

**Key Questions For Authors:**

1. Could the authors comment on how the sharpness of their instance-specific bounds differs from that of the global bounds of Gao et al. 2022, via either theoretical analysis or empirical comparison?
2. Could the authors explain more concretely what their instance-specific bounds may imply for a practitioner using causal discovery algorithms or a researcher developing them, especially when the true MEC is unknown?

**Limitations:**

Yes.

**Strengths And Weaknesses:**

We thank the authors for their valuable contribution.

A. Soundness

I. Strengths
1. The theoretical results are detailed, giving an extensive characterization of Fisher dimension for different types of graph structures and parameterizations. The proofs are well-written and parseable on a skim, though I have not thoroughly checked their correctness.

B. Presentation

I. Strengths
1. Proof sketches for theoretical results help understanding.
2. The geometric intuition for the results is useful in grasping the key idea underlying the Fisher dimension and lower bound.

II. Weaknesses
1. Certain terms could benefit from definitions, either in the main paper or in the appendix, e.g., Markov blanket, Fisher-Rao metric.
2. Detailed empirical results comparing other complexity proxies to the FD are not provided; only the best proxy is given in Table 3.
3. The authors do not detail how the empirical sample complexity is computed in Sec. 6.1.

C. Significance

I. Strengths
1. The problem of understanding the sample-complexity of causal discovery is well-motivated.
2. The authors characterize when the difficulty of causal discovery does not depend on aspects of graph structure (e.g., edge degree, node size, etc.) but rather on the parameters of the underlying SEM. This is a novel and surprising insight–it would be very helpful if the authors could discuss what this implies for future research into causal discovery algorithms.

II. Weaknesses
1. The authors do not discuss when/how the instance-dependent sample complexity bounds are sharper than the min-max bounds of Gao et al. 2022 (a work they cite). This is also not included as a baseline in the experiments.
2. The empirical results show that Markov blanket size is a better predictor of empirical sample complexity than Fisher dimension for Erdos-Renyi graphs. Fisher dimension is more informative for chains and trees. 3. 3. Since chains and trees are presumably less common than more general DAGs, this undermines the practical upshot of their bounds.
4. The task concerns the complexity of exactly learning the true MEC–in particular, distinguishing it from the nearest possible incorrect MEC. While this is an essential theoretical first step, PAC-style bounds parameterized in terms of learning an estimate MEC that is distance d from the true MEC for general d may be more informative in practice.
5. Computing instance-dependent bounds requires knowledge of the ground truth graph. The authors suggest a post-hoc analysis using the Fisher dimension of an MEC learned from data. However, it’s not clear what such an analysis tells you if the true MEC is far from the learned MEC. Even if the true MEC and learned MEC coincide, it’s not clear whether practitioners would benefit from a post-hoc analysis of the difficulty of the problem. Still, this does not limit the theoretical impact of the paper, and adds nuance to a common intuition people may have that the difficulty of structure learning lies in the true graph structure.

D. Originality

 I. Strengths
1. The authors introduce the first instance-dependent sample complexity bounds for causal discovery. Their theoretical analysis is technically sophisticated and substantially different from that of Gao et al 2022 (their nearest neighbor work).
2. The geometric intuition for when causal discovery is difficult is insightful and can potentially be generalized to other analyses of the sample complexity of causal discovery.

---

> ### Author Rebuttal · Authors · 2026-03-30
>
> We appreciate the reviewer for their thoughtful feedback and for recognizing the novelty of our instance-dependent sample complexity bounds.
>
> ## Response to Weaknesses
>
> **B. Representation**
>
> We appreciate the suggestions regarding the clarity and completeness of our presentation. We will incorporate formal definitions for the Markov blanket and the Fisher-Rao metric in the final version to ensure the paper is self-contained. Regarding Table 3, while we only reported the best-performing proxies to maintain brevity, we will include the full ranking in the appendix. Finally, we will clarify the methodology to calculate the empirical sample complexity in Section 6.1.
>
> **C. Significant**
>
> - We appreciate the point regarding the comparison with Gao et al. (2022); while their work establishes essential minimax structural rates, our Fisher Dimension (FD) captures instance-specific variation that global bounds miss. For example, two graphs with the same maximum in-degree $q=1$ can have vastly different complexities based on parameter scaling.
>
> - It is a fair point that our predictor is weaker for Erdos-Renyi (ER) graphs where structural heterogeneity dominates, and we will clarify that FD is most effective when structural features are approximately controlled.
>
> - We agree that extending this theoretical first step toward PAC-style bounds would be more informative for practical distance-based estimates; we appreciate this suggestion and will incorporate a discussion on moving beyond exact recovery as a key direction for future work.
>
> - Regarding the utility of post-hoc analysis, we appreciate the nuance provided; as discussed in Section 5.3, we view the estimated Fisher Dimension as a valuable diagnostic tool that helps practitioners assess the fragility of learned structures and understand why specific datasets present higher difficulty.
>
> ## Answering Questions
>
> **1. How does the sharpness of instance-specific bounds differ from global bounds?**
>
> The sharp distinction lies in global versus instance-specific difficulty. Global minimax rates, such as $q \log(d/q)$, characterize the worst-case difficulty for a class of graphs. In contrast, our Fisher Dimension captures variation within those classes; for instance, in a simple chain where $q=1$, the complexity can range from $O(1)$ to $O(d)$ depending on edge scaling, a variation that global bounds cannot explain.
>
> **2. What do these bounds imply for a practitioner when the true MEC is unknown?**
>
> For practitioners, our results offer a way to move beyond structural intuition by highlighting how parametric properties drive hardness. When the true MEC is unknown, researchers can run a first-pass discovery and compute the estimated FD (Section 5.3) as a post-hoc diagnostic, which is useful for understanding why a dataset was hard, comparing relative difficulty, and flagging fragile learned structures.

---

> > ### Author Rebuttal · Reviewer_yytz · 2026-04-03
> >
> > We appreciate the authors' clarifications and have one follow-up question regarding the post-hoc diagnostic (Q#2). Is the correctness of the learned MEC a necessary condition for the FD computed as such to be insightful?

---

> > > ### Author Response · Authors · 2026-04-08
> > >
> > > Not strictly. If the learned MEC is wrong, the Fisher dimension computed from it can still tell you something about the learned output itself, but it is no longer guaranteed to reflect the true difficulty of the underlying causal-discovery problem. Section 5.3 already frames it as a retrospective, post-hoc diagnostic, and explicitly warns that when the learned structure differs from the true one, the estimate may fail to reflect the true instance difficulty; it is most reliable when structure recovery is successful.

---

### Official Review · Reviewer_m4wB · 2026-03-12

**Soundness:** 3
**Presentation:** 3
**Significance:** 3
**Originality:** 3
**Overall Recommendation:** 5
**Confidence:** 3

**Summary:**

The paper studies instance-specific sample complexity for Markov equivalence class recovery in linear Gaussian structural equation models. The authors introduce the Fisher dimension, defined as the inverse squared minimum partial correlation that must be detected for MEC recovery. They prove it provides both a lower bound (for any algorithm) and an upper bound (for PC), matching up to logarithmic factors. They provide also an empirical validation on synthetic graphs demonstrates agreement between metric and causal discovey perfomance.

**Compliance With Llm Reviewing Policy:**

Affirmed.

**Final Justification:**

I had some concerns and questions regarding bound applicability and use cases. During rebuttal authors addressed my concerns and answered question. The paper could potentially use bit more experiments overall my questions were answered and I have a positive view of the paper.

**Key Questions For Authors:**

1. Given the log d gap between the upper and lower bounds, and the fact that the upper bound is established only for PC, is the Fisher dimension tight enough in practice to be used for comparing the sample efficiency of different causal discovery algorithms (e.g., PC vs GES vs penalized likelihood methods)?
2. Beyond characterizing instance difficulty, what other practical applications do the authors envision for the Fisher dimension?

**Limitations:**

yes

**Strengths And Weaknesses:**

# Strengths
- The paper provides matching lower and upper bounds (up to log d) for sample complexity of MEC recovery. The lower bound applies to any algorithm via Le Cam's method, and the upper bound is rigorously established for the PC algorithm.
- The curvature characterization conjecture (Conjecture 3.9) is proved specifically for chain graphs under weak-edge scaling (Theorem 3.11).
- Exact scaling F([G]) = Θ(1) is proved for complete DAGs (Proposition 4.4), providing a concrete non-trivial example where the Fisher dimension is fully characterized.
- The theoretical predictions are validated empirically, showing strong consistency between F([G]) · log d and observed sample complexity for chains and trees , demonstrating that the framework captures parametric difficulty effectively for structurally homogeneous graph families.

# Weaknesses

Weaknesses

* The empirical results for Erdős–Rényi graphs are not very convincing, especially for low and medium F showing little correlation.
* The Fisher dimension captures only parametric difficulty, as demonstrated by the strong results on trees and chains where graph structure is held constant across instances.
* No experiments are reported for other causal discovery methods for example based on score such as GES. This limits the claim that the Fisher dimension is an algorithm-independent complexity measure, since only the lower bound (via Le Cam) applies generally while the upper bound and all empirical evidence are PC-specific. It could be useful in showing that the dimension can be useful.
* The random graph experiments only consider Erdős–Rényi graphs. Scale-free graphs, which are popular in applied causal discovery, are not evaluated despite potentially exhibiting different faithfulness and partial correlation behavior due to their hub structure.
* The experimental scale is limited to d ≤ 20, which is quite small. Near-faithfulness issues and scaling behavior may not fully manifest at this size, making it difficult to assess whether the theoretical predictions hold in more realistic settings.

---

> ### Author Rebuttal · Authors · 2026-03-30
>
> We are grateful the reviewer for their thoughtful feedback and for recognizing the technical solidity and potential impact of our work.
>
> ## Response to Weaknesses
>
> - We appreciate the observation regarding the Erdős-Rényi (ER) results and acknowledge that the Fisher dimension (FD) shows a weaker correlation in this setting compared to trees or chains. This is because FD is designed to track parametric difficulty whereas ER families exhibit substantial structural variation across instances. As noted in Table 3, structural proxies like average Markov blanket size can be more predictive than FD when structural heterogeneity dominates (0.58 vs. 0.37). We plan to explore hybrid metrics that better integrate both structural and parametric variance in future work.
>
> - We agree that the current framework focuses primarily on parametric difficulty. As the reviewer noted, our results on trees and chains where the structure is held constant demonstrate this strength. This remains an open problem, and we plan to investigate incorporating structure-dependent complexity in future work.
>
> - The FD is intrinsically algorithm-independent by definition (Def. 3.1). It characterizes properties of the underlying instance, not of any specific method. We confirm that it may be easy to conflate the FD with the empirical number of samples required, which is both method-dependent and instance-dependent; therefore, we will revise the manuscript to clarify this distinction. We also appreciate the suggestion to include other algorithms like GES. We did not include experiments with other methods such as GES because we have not yet derived an upper bound on their sample complexity in terms of the FD.
>
> - The point regarding scale-free graphs is well-taken. Such graphs often feature hub nodes with high in-degrees, which can significantly impact the conditioning sets used in partial correlation tests. While our ER experiments provided an initial look at random structures, we agree that scale-free models represent an important class of real-world networks, and we will look to incorporate them in future.
>
> - We acknowledge that the experimental scale is limited to $d \le 20$. This constraint was primarily driven by the computational cost of the scaling experiments and the fact that the empirical sample complexity for ER graphs hit a censoring ceiling of $n^* = 10$K at $d=20$. Despite the small size, the observed growth in sample complexity correlates strongly with $\log d$ ($r > 0.92$), which is consistent with our theoretical predictions. We will aim to use more efficient implementations to test higher dimensions in future studies.
>
>
> ## Answering Questions
>
> **1. Is the Fisher dimension tight enough in practice to compare different algorithms?**
>
> The FD serves as a robust common baseline for hardness because our lower bound is information-theoretic and applies to any algorithm. While the matching upper bound is currently established specifically for the PC algorithm , the theory demonstrates that FD captures critical instance-specific variation that global minimax bounds miss. For example, two chains with the same maximum in-degree $q=1$ can have an FD ranging from a constant $2$ to a size-dependent $d+1$, depending purely on the parametric edge scaling. This suggests that while FD is not yet a universal cross-algorithm ranking tool, it provides the necessary granularity to identify when an algorithm's performance is limited by the inherent flatness of a specific causal manifold rather than structural sparsity alone.
>
> **2. What other practical applications do you envision for the Fisher dimension?**
>
> Beyond characterizing difficulty, we envision the FD being used for post-hoc reliability diagnostics. For example, after running a discovery algorithm, a researcher could estimate $\hat{\mathcal{F}}([\hat{G}])$ to assess how close the results were to the limits of the data's signal. High FD would serve as a warning that the learned structure was barely distinguishable from other MECs, suggesting the need for more data or interventional experiments.

---

> > ### Author Rebuttal · Reviewer_m4wB · 2026-04-03
> >
> > I would like to thank the reviewers for their answers and clarification. I would like to keep the score and the  overall positive assessment of the work.

---

### Official Review · Reviewer_QE3K · 2026-03-13

**Soundness:** 3
**Presentation:** 2
**Significance:** 2
**Originality:** 3
**Overall Recommendation:** 2
**Confidence:** 3

**Summary:**

This paper proposes the Fisher dimension, an instance-dependent complexity measure for causal discovery in linear Gaussian SEMs, defined by the smallest “critical” nonzero partial correlation needed to distinguish a Markov equivalence class (MEC). It proves nearly tight (up to logs) sample complexity bounds showing MEC recovery requires and is achieved with $n$ scaling like $F([G])$, and provides algorithms plus synthetic experiments indicating $F([G])\log d$ predicts when PC succeeds better than common structural proxies in parameter-driven regimes.

**Compliance With Llm Reviewing Policy:**

Affirmed.

**Final Justification:**

I have concerns about the paper’s presentation and the rebuttal engagement. I would like to see a more complete, rigorous, and carefully written version of this work, but for now I keep my original score.

**Key Questions For Authors:**

1. Could you please list several practical applications/usages of such kind of instance-based complexity?
2. Can you clarify to what extent Fisher dimension is intended as a universal instance-dependent complexity measure (across diverse graph families), and whether you see a principled way to incorporate structural difficulty into the theory/metric (e.g., an augmented complexity parameter that combines $\rho_{\min}$ with a structural term), supported by additional experiments beyond chain/tree (e.g., broader ER regimes, scale-free/small-world, varying MEC size or MB size)?

**Limitations:**

yes

**Strengths And Weaknesses:**

**Strength:**

The paper makes a clear theoretical contribution with a principled instance-dependent characterization of sample complexity, supported by nearly matching upper and lower bounds. The metric it proposes is simple and interpretable, and the overall argument is cohesive by combining algorithmic analysis with information-theoretic reasoning.

**Weakness：**

1. Insufficient rigor. Section 2.4 mixes up the PC algorithm with one specific CI test: PC is a CI-test–based framework, and it only turns into “partial-correlation testing” after you assume linear-Gaussian data and pick something like Fisher’s z. As written, it’s easy to walk away unsure what comes from PC itself versus what comes from the particular test and distributional assumptions. Also, if you describe PC as “skeleton discovery + orientation,” please explicitly cite Meek’s rules for the orientation step.
2. Missing direvation: Remark 3.2 gives a specific curvature scaling $(1-\rho^2)^{-2}$ under the Fisher–Rao metric, but I could not find a derivation or a reference in the appendix; adding a short derivation (even for a 2D correlation submanifold) or a citation would make the information-geometric interpretation more convincing.
3. Empirically, its predictive power is strong on relatively homogeneous families (chain/tree) but becomes noticeably weaker on ER graphs, as mentioned by the authors, where simple structural proxies (e.g., Markov blanket size) can be more predictive. I think this significantly narrows the applicable settings of the method.
4. Computing $F([G])$ relies on (near-)accurate partial correlations, yet in realistic settings one only has finite samples—precisely when the quantity is most needed. With small $n$,  $\widehat{\rho}_{\min} $ (hence $\widehat{F}$ ) can be highly unstable, leading to unreliable “required $n$” estimates. To my understanding, to make the metric actionable for data collection, one would need an explicit sequential protocol (e.g., confidence bounds / stopping rules) of repeatedly collecting more data according to the new estimation of Fisher Dimension, and a discussion of how $\widehat{F}$ behaves in the low-sample regime; these aspects are not fully developed in the paper.
5. Formatting of Mathematical Equations: The authors systematically omit punctuation marks at the end of display equations throughout the entire manuscript. In standard academic typesetting, equations are grammatically part of the sentence and must be punctuated accordingly (e.g., ending with a period or a comma where appropriate). Please meticulously revise the manuscript to ensure all equations follow standard mathematical typesetting conventions to ensure professionalism.

---

> ### Author Rebuttal · Authors · 2026-03-30
>
> We thank the reviewer for the careful reading and constructive suggestions.
>
> ## Response to Weaknesses
>
> **1. PC algorithm vs. partial-correlation/Fisher-$(z)$ testing**
>
> We agree. Section 2.4 currently describes PC directly through partial-correlation tests, while Theorem 3.4 / Appendix A instantiate the upper bound with Fisher-$(z)$-based partial-correlation CI tests under linear-Gaussian assumptions. The wording in Section 2.4 and in the theorem statement should distinguish these layers: PC as a generic CI-testing framework, and the achievability result as PC instantiated with Fisher-$(z)$ / partial-correlation CI tests in the linear-Gaussian setting. We should also cite Meek's orientation rules and describe the orientation step as unshielded-collider orientation followed by Meek-rule propagation. Within the linear-Gaussian setting, the Fisher-dimension definition and the lower bound are not tied to a particular CI-testing algorithm; the upper bound is proved by instantiating PC with Fisher-$(z)$ / partial-correlation CI tests.
>
> **2.Missing derivation for Remark 3.2**
>
> Thank you, we concur and will add a derivation for Remark 3.2. Standardizing the conditional covariance to the bivariate family $\Sigma(\rho) = \begin{pmatrix} 1 & \rho \\\\ \rho & 1 \end{pmatrix}$ yields the Fisher-Rao metric coefficient:
>
> $$g_{\rho\rho} = \frac{1}{2} \operatorname{tr}\!\Big(\Sigma^{-1} \partial_\rho \Sigma \;\Sigma^{-1} \partial_\rho \Sigma\Big) = \frac{1 + \rho^2}{(1 - \rho^2)^2}$$
>
> This scales as $(1 - \rho^2)^{-2}$ given the bounded factor $1 + \rho^2 \in [1, 2]$. Consequently:
>
> $$\mathrm{KL}(P_\rho \,\|\, P_{\rho+h}) = \frac{1}{2} \cdot \frac{1 + \rho^2}{(1 - \rho^2)^2} \, h^2 + O(h^3)$$
>
> **3. Weaker predictive power on ER graphs**
>
> We acknowledge that FD isolates parametric detectability, excelling in structured families (trees/chains) over heterogeneous ones like ER. Our results reflect this: Spearman correlations reach 0.91–0.94 for trees/chains but only 0.66 for ER, where Markov blanket size is more predictive. We will specify that FD captures signal within fixed graph types. A principled future extension involves replacing $O(\log d)$ with a graph-dependent $N_{\text{eff}}([G])$ to better quantify structural search complexity. We regard this as a concrete open direction and will frame it as such in the text.
>
> **4.Finite-sample estimation**
>
> Indeed, Section 5.3 is a post-hoc diagnostic, not a sequential oracle. Our theory (Lemma 5.2, Prop 5.3) establishes fixed-$n$ convergence for equivalence classes with $\rho_{\min} > 0$. We will clarify this scope, noting Experiments 2-4 utilize population covariance and bypass low-$n$ $\widehat{F}$ instability. This instability arises from the $1/\rho^2$ singularity at zero. Our $O(\sqrt{\log(d/\delta)/n})$ bounds are fixed-$n$ and not anytime-valid; developing time-uniform confidence sequences for sequential stopping is future work, not a present claim.
>
> **5. Equation punctuation**
>
> We agree and will address this throughout the manuscript.
>
> ## Answering Questions
>
> **1. Practical applications of instance-based complexity**
>
> Several practical uses include: conservative sample-size planning from sufficiently informative pilot data, where one estimates $F([G])$ to inform how many additional samples are needed rather than relying on worst-case minimax formulas; post-hoc diagnosis of why a learned CPDAG was easy or hard, and whether the difficulty was parametric (weak signals) or structural; identifying fragile edges/orientations associated with the smallest relevant partial correlations, which may warrant targeted intervention or more conservative interpretation; adaptive data-collection heuristics based on confidence intervals for $\rho_{\min}$ (a potential future use, not yet fully developed in the present paper); and benchmarking, where FD can stratify instances by parametric hardness rather than only by coarse structural statistics such as $d$ or density.
>
> **2. Universality and structural augmentation**
>
> FD is a parametric, instance-dependent term for linear-Gaussian MECs, not a universal scalar for all graph families. It answers the specific question: how hard is the weakest nonzero partial correlation that must be detected? This is the dominant difficulty axis when structural features are controlled, but it does not capture structural search difficulty.
>
> To address this, a principled refinement would replace the coarse $O(\log d)$ union-bound term with a graph-dependent $\log N_{\text{eff}}([G])$, representing the effective CI test search space. This aligns with our current analysis, which already explicitly counts PC CI tests. Empirically, Table 3 suggests that Markov blanket size significantly impacts heterogeneous families. Since we lack the matching theory for a specific augmented complexity formula, we will present this as a concrete open direction and soften any claims of FD’s universality.

---

> > ### Author Rebuttal · Reviewer_QE3K · 2026-04-04
> >
> > Thank you for the detailed rebuttal. My questions are largely addressed by the planned revisions, so I mark them as resolved. That said, the clarifications in the rebuttal also make the scope of the contribution appear narrower than I initially understood, and I still believe the overall impact is limited for this venue. Therefore, I will keep my score unchanged.

---

### Decision · Program_Chairs · 2026-04-30

**Decision:**

Accept (regular)

**Comment:**

Paper introduces Fisher dimension (defined as inverse squared minimum conditional partial correlated for detecting the weakest edge in a graph) for linear Gaussian causal models (SEMs) and show using LeCam's method that O(Fisher dimensions) samples are necessary and union bounding CI tests with required power to establish a certain graph structure with max in degree , one gets a sample complexity that is O(Fisher dimensions log d). Authors show a curvature related interpretation of Fisher dimension.

Well conditioned SEMs (O(1) edge weights, finite variances) have O(1) fisher dimension irrespective of graph structure. This also seems an interesting result.

Concerns reviewers raised:
1) Avg Markov Blanket size seems to be a good proxy metric for Erdos Renyi graphs while fisher dimension seems to predict sample complexity for chains and trees

2) The result does not  prove a sequential testing related "anytime n" guarantee but uses union bounds to establish a "required n" style result

These authors do acknowledge in their rebuttal. 2) issue which is also related controlling False discovery rates of PC algorithm itself is a non trivial exercise taking into account sequential nature of testing at finite
"given n " samples. Most interesting causal graphs are also sparse - (tree like).

Therefore, on balance although based on simple LeCam and union bound based testing results, curvature based interpretation and structure independent results (in well conditioned SEMs) make this a good contribution.